# Anterior Colporrhaphy and Paravaginal Repair for Anterior Compartment Prolapse: A Review

**DOI:** 10.3390/medicina60111865

**Published:** 2024-11-14

**Authors:** Wing Lam Tsui, Dah-Ching Ding

**Affiliations:** 1Department of Obstetrics and Gynecology, Hualien Tzu Chi Hospital, Buddhist Tzu Chi Medical Foundation, Tzu Chi University, Hualien 970, Taiwan; karenwinglam0505@tzuchi.com.tw; 2Institute of Medical Sciences, Tzu Chi University, Hualien 970, Taiwan

**Keywords:** anterior compartment, pelvic organ prolapse, anterior colporrhaphy, paravaginal repair, surgery

## Abstract

Pelvic organ prolapse, particularly in the anterior compartment, is a prevalent condition that significantly impacts women’s quality of life. Two common surgical approaches for managing anterior vaginal wall prolapse are anterior colporrhaphy and paravaginal repair. Anterior colporrhaphy, a traditional technique, involves the plication of weakened fascial tissues to restore support to the bladder and anterior vaginal wall. Paravaginal repair addresses lateral detachment of the anterior vaginal wall by reattaching it to its supportive structures. This review aimed to compare the indications, techniques, and outcomes between these surgical methods, discussing their efficacy, recurrence rates, and complications. Although anterior colporrhaphy is widely used, paravaginal repair may offer superior results in specific cases, particularly those involving lateral defects. The review also explored the evolution of these techniques, the role of grafts and mesh, and the potential benefits of minimally invasive approaches such as laparoscopy and robotic surgery. The goal is to provide clinicians with comprehensive insights into choosing the appropriate surgical option based on individual patient anatomy and clinical presentation, thus optimizing outcomes and minimizing recurrence.

## 1. Introduction

Pelvic organ prolapse (POP) is a condition where the pelvic organs, such as the bladder, uterus, or rectum, descend into or outside the vaginal canal because of weakened pelvic floor support [1]. It is a prevalent disorder, especially among women who have undergone childbirth, experienced menopause, or advanced age [2]. POP can cause discomfort, urinary or bowel dysfunction, and sexual issues, thus significantly affecting a woman’s quality of life, [3].

Pelvic organ prolapse (POP) is a prevalent gynecological condition, particularly in older women. Prevalence rates vary across studies, with estimates ranging from 10.3% in rural Pakistan [4] to 40% globally [5]. However, a South Korean study reported a much lower prevalence of 180 per 100,000 women over 50 [6]. Risk factors for POP include increasing age, parity, body mass index, and fetal macrosomia [5]. Constipation was also found to significantly increase POP risk [6]. POP significantly impacts women’s quality of life, with 60.8% of affected women in rural Pakistan reporting moderate to great effects [4]. Treatment options include pessaries and surgery, with surgery peaking around age 70 and pessary use increasing dramatically after age 65 [6]. As the elderly population grows, the demand for POP treatment is expected to increase [7].

The pelvic floor is a complex structure composed of muscles, ligaments, and connective tissue that spans the base of the pelvis [8]. It consists primarily of two muscle groups: the levator ani (which includes the pubococcygeus, puborectalis, and iliococcygeus muscles) and the coccygeus muscle [9]. These muscles form a supportive sling from the pubic bone at the front to the coccyx at the back, encircling the openings of the urethra, vagina, and rectum. In addition to these muscles, strong connective tissues, such as the endopelvic fascia, contribute to the overall support structure [10]. The pelvic floor maintains the position of critical pelvic organs—such as the bladder, uterus, and rectum—while allowing for functions like urination, defecation, and childbirth [11]. It also plays a vital role in continence and sexual function by controlling the contraction and relaxation of pelvic openings [12]. Proper coordination of the pelvic floor is essential to maintain organ stability and prevent conditions like pelvic organ prolapse.

POP is classified by different compartment types based on the specific organ and area affected [13]. Anterior compartment prolapse, or cystocele, is the most common type and involves the descent of the bladder and anterior vaginal wall [14]. Posterior compartment prolapse, or rectocele, refers to the descent of the rectum, while apical compartment prolapse occurs when the uterus or vaginal vault (after hysterectomy) descends [14]. These classifications help in diagnosing and managing POP, as each compartment presents distinct symptoms and requires different treatment approaches [15].

Common causes of anterior compartment prolapse include childbirth trauma, particularly vaginal deliveries; connective tissue disorders; obesity; and chronic increased intra-abdominal pressure (e.g., from chronic coughing or heavy lifting) [16]. Symptoms of anterior compartment prolapse often include a sensation of bulging or pressure in the vaginal area, which patients frequently describe as a feeling of something “falling out” [17]. Urinary symptoms are common and may manifest as stress urinary incontinence, urgency, or difficulty emptying the bladder [18]. Additionally, recurrent urinary tract infections can occur due to incomplete bladder emptying [19]. Urinary incontinence can lead to embarrassment, social withdrawal, and reduced physical activity due to the fear of leakage, while recurrent infections cause discomfort and frequent medical interventions [18,19]. Discomfort or pain during sexual intercourse is another notable symptom, impacting the quality of life for affected individuals [20]. Sexual dysfunction, often stemming from pain or changes in vaginal anatomy, can strain intimate relationships and diminish overall quality of life [20]. These symptoms can vary in severity, depending on the degree of prolapse, and often prompt individuals to seek medical evaluation and treatment.

The severity of anterior compartment prolapse is graded using the Pelvic Organ Prolapse Quantification system [21], which measures the descent of pelvic organs relative to the hymen. Prolapse may range from mild (stage 1) to complete (stage 4) [22].

Anterior compartment prolapse is the most common type of pelvic organ POP, occurring in 54–79% of women undergoing POP surgery [23,24]. It is often associated with apical defects, especially in older women. Surgical intervention rates for POP vary significantly across countries, with a median rate of 1.38 per 1000 women [24]. Native tissue repair remains a common approach, although it is associated with higher rates of prolapse recurrence compared to mesh repairs [25]. The choice of surgical technique remains controversial, with no clear consensus on the optimal approach [26]. Factors such as age and previous hysterectomy may influence the risk of specific compartment involvement and should be considered when planning surgical interventions [23].

The management of anterior compartment prolapse ranges from conservative treatments, such as pelvic floor muscle training and pessary use, to surgical intervention [14,27]. When conservative remedies fail to alleviate these symptoms, surgical intervention may become necessary [28]. POP can worsen over time if not effectively treated. In cases where the prolapse is severe or continues to progress despite non-surgical interventions, surgery offers a more permanent solution by directly addressing the underlying anatomical defects [29]. Surgical repair options include anterior colporrhaphy and paravaginal repair, each of which addresses the structural defects causing the prolapse [29]. Studies comparing these methods have shown mixed results regarding recurrence rates. One study found standard plicating and rolling anterior colporrhaphy techniques to be equally effective, while the purse-string technique had higher recurrence rates [30]. Paravaginal repairs, while technically more difficult, may offer better outcomes [31]. Another study reported lower anatomic failure rates with mesh-reinforced paravaginal repair compared to colporrhaphy, but similar composite failure rates and quality of life improvements [32]. However, mesh-related complications remain a concern [33]. A recent retrospective study found comparable surgical outcomes between anterior repair and paravaginal repair at 1–2 months, 3–6 months, and 1 year post-surgery, although paravaginal repair had longer operation times [29]. While mesh-reinforced repairs show promising short-term success, long-term durability and safety require further research [33]. Additionally, surgical intervention can help prevent complications such as further prolapse, chronic infections, or worsening incontinence, which could arise if the condition remains untreated [28].

In cases where the pelvic floor’s supportive structures weaken, leading to anterior prolapse, surgical management becomes crucial, with anterior colporrhaphy and paravaginal repair serving as key approaches. These procedures are vital in restoring the anatomical integrity of the pelvic organs, alleviating symptoms, and improving the quality of life for patients by addressing both central and lateral defects in the anterior vaginal wall.

### Inclusion and Exclusion Criteria

The review was systematically searched with the keywords “anterior colporrhaphy, paravaginal repair, transvaginal mesh, pelvic organ prolapse, risk factor, recurrence” from their respective inception to 31 August 2024. Synonyms and derivatives of keywords were also used. The bibliographies of relevant reviews and included studies were also scrutinized. Appendix A describes the search strategy used for the PubMed database.

Case reports, case series, reviews, and animal or laboratory studies were excluded.

## 2. Anatomy of the Anterior Vaginal Wall and Pelvic Support

### 2.1. Anatomy of the Anterior Compartment

The anterior compartment of the pelvis primarily consists of the bladder, urethra, and the anterior vaginal wall [34]. The anterior vaginal wall lies adjacent to the bladder and urethra, providing structural support to these organs [35]. The pubocervical fascia, a fibrous connective tissue, is an essential structure in maintaining the integrity of the anterior compartment by attaching the bladder to the anterior vaginal wall [36]. Defects or weakening of this fascia significantly contribute to the development of anterior compartment prolapse.

### 2.2. Role of Pelvic Ligaments, Fascia, and Muscles

Key structures of the anterior compartment include the endopelvic fascia, cardinal and uterosacral ligaments, levator ani muscles, and pubocervical fascia. The endopelvic fascia provides a connective tissue framework that supports the pelvic organs [37]. The cardinal and uterosacral ligaments provide vertical support to the pelvic organs and are crucial for maintaining the position of the bladder and uterus [38]. The levator ani muscles form the pelvic floor and provide essential support for the pelvic organs. This group of muscles including the puborectalis, pubococcygeus, and iliococcygeus plays a vital role in resisting the descent of the pelvic organs under pressure [8]. The pubocervical fascia specifically supports the bladder and anterior vaginal wall, preventing anterior compartment prolapse. Damage to any of these structures, often through childbirth, aging, or surgery, can weaken the pelvic support system and lead to prolapse [10].

### 2.3. Pathophysiology of Anterior Vaginal Wall Prolapse

Factors contributing to anterior compartment prolapse include childbirth trauma, aging, chronic strain, and estrogen deficiency. For childbirth trauma, vaginal deliveries, especially with large babies or prolonged labor, can overstretch or tear pelvic muscles and fascia [2,39]. With respect to aging, connective tissue loses elasticity over time, reducing the ability of ligaments and fascia to support pelvic organs [40]. For chronic strain, conditions such as chronic coughing, constipation, and heavy lifting can increase intra-abdominal pressure, contributing to pelvic organ prolapse [2]. Finally, in postmenopausal women, lower estrogen levels lead to the atrophy of vaginal and pelvic tissues, making them more prone to prolapse [41]. The combination of these factors leads to a progressive loss of structural support, resulting in the prolapse of the anterior vaginal wall and bladder.

## 3. Anterior Colporrhaphy

### 3.1. Definition

Anterior colporrhaphy is a traditional surgical technique used to repair prolapse of the anterior vaginal wall [25]. It involves strengthening the anterior compartment by plicating or suturing the weakened fascia beneath the vaginal wall, aiming to restore pelvic support [42].

### 3.2. Indications

The procedure is commonly performed when conservative measures such as pelvic floor exercises or pessary use fail, and patients experience symptoms such as vaginal bulging, urinary difficulties, and discomfort during intercourse [42].

### 3.3. Surgical Technique

The surgical technique begins with a vaginal incision to expose the prolapsed bladder, followed by plication of the pubocervical fascia and closure of the incision [43]. Some surgeons modify the procedure by incorporating graft materials to provide additional support in cases of severe tissue weakness. This variation can improve the longevity of the repair, although complications such as graft erosion and infection may occur.

### 3.4. Mechanisms

Plicating or suturing the weakened fascia during procedures like anterior colporrhaphy restores the anteroposterior support of the bladder by reinforcing the connective tissue layer that lies between the bladder and the vaginal wall [44]. The fascia, which normally acts as a supportive structure, can become stretched or weakened due to factors like childbirth, aging, or increased intra-abdominal pressure. This weakening allows the bladder to descend, leading to prolapse [9].

By plicating or tightening the fascia with sutures, the surgeon strengthens this layer and pulls the bladder back into its normal position [45]. This re-establishes tension and support along the anteroposterior axis, which runs from the front to the back of the body [45]. Restoring this support prevents the bladder from sagging into the vaginal wall and improves its functional stability, thus relieving symptoms like urinary incontinence and reducing the risk of future prolapse [45]

### 3.5. Why Surgery Needed

Surgery is often required for pelvic organ prolapse when conservative treatments fail to provide adequate relief or the prolapse becomes too severe [46]. While non-invasive methods like pessaries, pelvic floor exercises, and lifestyle modifications can help manage mild to moderate cases, they do not correct the underlying structural defects causing the prolapse [47]. In cases of more advanced prolapse, these conservative measures may only offer temporary symptom relief and may not prevent further progression of the condition [46].

For second- or third-degree prolapse, patients are often quite uncomfortable and desire therapy [48]. This suggests that conservative treatments may be less effective or insufficient for these more severe cases [49]. The need for surgical intervention in more severe cases suggests that conservative treatments like pessaries may be less effective in managing the symptoms and complications associated with advanced prolapse [50]. Surgical intervention becomes a consideration for more severe cases [51].

### 3.6. Outcomes

The success rates of anterior colporrhaphy are generally favorable in the short-term, with initial relief of prolapse symptoms seen in 70% to 90% of patients [46,52]. However, long-term durability remains a challenge, as recurrence rates can range from 20% to 40% [53,54]. The recurrence of POP after surgery is influenced by several factors including age, degree of prolapse, and postoperative tissue quality. Older patients are at higher risk of recurrence due to the natural weakening of pelvic tissues [16]. The degree of prolapse is a significant factor, with more severe or advanced prolapse associated with a greater likelihood of recurrence because of extensive tissue damage and increased strain on the repair [55]. Postoperative tissue quality, influenced by factors such as age, genetics, and conditions like diabetes or smoking, also plays a key role in determining the durability of the repair [56]. Other contributors include obesity, chronic strain from activities like heavy lifting, and previous prolapse surgeries [57]. The choice of surgical technique as well as the use of grafts or mesh can improve outcomes but also present specific risks. Recognizing and managing these factors is crucial to minimize the risk of recurrence and ensure long-term success.

### 3.7. Long-Term Effectiveness

Anterior colporrhaphy remains a viable option for treating cystocele, with long-term success rates varying based on patient factors and surgical technique. For high-grade cystoceles, anterior colporrhaphy combined with pubovaginal sling reinforcement showed an 84.8% recurrence-free survival at 5 years [58]. However, anterior colporrhaphy alone may be less effective for severe or recurrent cases, with recurrence rates up to 78.9% after 5 years [59]. Reinforcing anterior colporrhaphy with polypropylene mesh for severe cystoceles demonstrated improved outcomes, with 75.7% of patients cured at 14–19 months follow-up [60]. Patient-specific factors influencing long-term effectiveness include cystocele severity, previous prolapse surgeries, hypoestrogenism, weight gain, and chronic coughing or constipation [59,61]. These findings emphasize the importance of tailoring surgical approaches based on individual patient characteristics and careful preoperative assessment to optimize long-term outcomes.

### 3.8. Complications

Complications following anterior colporrhaphy can include recurrence of prolapse, where cystocele may redevelop because of the ongoing weakening of pelvic tissues despite initial success [62].

Postoperatively, patients may develop different types of urinary incontinence that can significantly impact their long-term quality of life [63]. Stress urinary incontinence (SUI), the most common, occurs when physical activities like coughing, sneezing, or exercising put pressure on the bladder, leading to involuntary leakage, often due to changes in pelvic floor support following surgery [64]. Urgency incontinence, or overactive bladder, involves a sudden, intense need to urinate, often accompanied by leakage, which may result from nerve or muscle alterations post-surgery [65]. Mixed incontinence, combining both stress and urgency symptoms, can be particularly challenging as patients experience leakage during both physical exertion and sudden urges [66]. These types of incontinence can lead to discomfort, embarrassment, and social isolation, significantly reducing the patients’ ability to engage in daily activities and affecting their emotional well-being. Early interventions including pelvic floor therapy, medications, or secondary surgical procedures are essential to mitigate these complications and improve quality of life [67].

Another potential complication is dyspareunia, or pain during sexual intercourse, although most patients typically report improved sexual function after the procedure [45,68]. These complications reflect the challenge of achieving both functional and structural success in anterior colporrhaphy.

The postoperative recovery phase following POP surgery is crucial for long-term success and requires comprehensive patient education and structured care [69]. Patients must be informed about the importance of avoiding activities that could strain the pelvic floor, such as heavy lifting or high-impact exercise, and provided with guidance on wound care, recognizing complications and understanding the recovery timeline [70]. Pelvic floor rehabilitation is vital in this phase, helping restore muscle strength and improve bladder control through targeted exercises, often with the assistance of a physical therapist, to prevent complications like urinary incontinence or prolapse recurrence [71]. Regular follow-up care including pelvic exams and symptom assessments allows for the early detection of complications and ensures that the repair remains intact. By combining effective patient education, rehabilitation, and consistent follow-up, healthcare providers can improve surgical outcomes and enhance the patients’ quality of life after surgery [72].

### 3.9. Modifications

Several modifications of the procedure, such as more sophisticated plication techniques and the use of synthetic or biological grafts, aim to reduce the recurrence rate and enhance repair strength [44].

Graft materials used in these procedures can be either biologic or synthetic. Biologic grafts, derived from human or animal tissue, integrate more naturally into the body and are less likely to cause an immune reaction [73]. They are typically associated with fewer complications such as erosion or infection [73]. However, biologic grafts may not provide as durable or long-lasting support compared to synthetic options, and there is a risk of degradation over time [74]. Synthetic grafts, such as polypropylene mesh, offer stronger, more permanent reinforcement [75]. These are particularly beneficial in high-risk patients or those with recurrent prolapse. However, synthetic materials carry a higher risk of complications including mesh erosion, infection, and pain, which can sometimes require further surgical intervention [76,77]. Balancing these benefits and risks is critical in patient selection, and the choice of graft material should be tailored to the individual’s specific needs and anatomy.

The incorporation of these innovations, however, may introduce risks such as graft-related complications [78,79]. In April 2019, the U.S. Food and Drug Administration officially prohibited the sale and distribution of transvaginal mesh for POP repair [80]. This decision came after years of growing concerns regarding the safety and effectiveness of these devices, which had been associated with serious complications [80].

Despite these challenges, anterior colporrhaphy remains a widely performed and effective treatment option for patients with symptomatic anterior compartment prolapse.

## 4. Paravaginal Repair

### 4.1. Definition

Paravaginal repair is a surgical procedure designed to address the lateral detachment or defect in the vaginal wall, which is often linked to pelvic organ prolapse [81]. This procedure is aimed to correct the support structures of the vagina that may be compromised, leading to prolapse or other functional issues.

### 4.2. Indications

Paravaginal repair is indicated for patients with significant lateral detachment of the vaginal wall due to pelvic organ prolapse, especially when symptoms such as vaginal bulging, urinary incontinence, or discomfort persist despite conservative management. It is also considered in cases of recurrent prolapse that have not responded to previous treatments [82].

### 4.3. Surgical Techniques

The surgical techniques for paravaginal repair include the open abdominal approach and laparoscopic and robotic techniques. The open abdominal approach involves a large incision to access and repair the defect, offering direct visualization and access but with longer recovery times [81]. Alternatively, laparoscopic and robotic techniques utilize minimally invasive methods with small incisions, providing reduced recovery times and less postoperative pain, although they require specialized training and equipment [29,83,84]. The transvaginal approach, performed through the vaginal canal, allows for direct repair of the lateral detachment with typically shorter recovery times, but may offer limited visibility compared to that via abdominal methods [82,85].

This technique involves reattaching the vaginal wall to the arcus tendineous fascia pelvis (ATFP), a fibrous structure that extends from the pubic bone to the ischial spine, thereby restoring support to the anterior vaginal wall [86]. The pubocervical fascia, located between the bladder and the anterior vaginal wall, is crucial for maintaining the proper positioning of these structures and preventing anterior compartment prolapse such as cystocele [10]. By anchoring the reattached vaginal wall to the ATFP, the paravaginal repair effectively re-establishes the anatomical relationship between the vaginal wall and surrounding structures, offering a durable solution for patients with lateral vaginal wall defects and improving overall pelvic support [87].

### 4.4. Outcomes

Paravaginal repair is generally effective in correcting lateral defects, with a high rate of anatomical success and improvement in symptoms related to pelvic organ prolapse [83]. Long-term outcomes often show sustained enhancement in quality of life and a reduction in prolapse symptoms, although results can vary based on the technique used and the complexity of the condition [29].

### 4.5. Complications

Potential complications of paravaginal repair include the risk of injury to surrounding structures such as the bladder, urethra, or rectum, which can lead to additional functional issues [83,88]. There is also the possibility of prolapse recurrence, which might require further intervention or ongoing management. Other risks include postoperative infections, wound healing issues, and persistent pain or discomfort. Although they typically resolve over time, some can occasionally persist.

## 5. Comparison Between Anterior Colporrhaphy and Paravaginal Repair

### 5.1. Indications

Anterior colporrhaphy is generally indicated for patients with anterior vaginal wall prolapse [42]. It is suitable for those with uncomplicated anterior wall defects and is often chosen when there is a need to strengthen the vaginal wall and restore its anatomical support [89]. Paravaginal repair is indicated for more complex cases involving lateral detachment of the vaginal wall, often associated with pelvic organ prolapse [82]. It is particularly suitable for patients with significant lateral defects or recurrent prolapse or those who have not responded well to conservative treatments [82]. Paravaginal repair is often chosen when there is a need to address specific support structures beyond the anterior vaginal wall [90].

### 5.2. Surgical Techniques: Advantages, Disadvantages, and Outcomes

Anterior colporrhaphy and paravaginal repair are surgical techniques for treating anterior vaginal wall prolapse, but their standardization and outcomes vary widely [42]. Anterior colporrhaphy has been the standard surgical treatment for anterior vaginal prolapse for a long time [91] and offers simplicity, ease of approach, and shorter operation time. However, it has a high recurrence rate and potential for vaginal shortening. Laparoscopic paravaginal repair is considered a more anatomic repair of lateral defects [83]. It offers advantages like the preservation of vaginal length, is minimally invasive, improved visualization, reduced bleeding risk, and faster recovery [83]. However, it has a longer operation time, technical complexity, and potential for complications [29].

A comparison study found that both anterior colporrhaphy and paravaginal repair significantly improved prolapse symptoms, with comparable outcomes at 1-year post-surgery, though paravaginal repair had longer operation times [29]. Anterior colporrhaphy is associated with high recurrence rates, leading to the exploration of mesh repairs [91]. However, mesh use in anterior colporrhaphy resulted in more complications without significant improvement in outcomes compared to traditional anterior colporrhaphy [91]. Laparoscopic paravaginal repair offers advantages such as improved visualization, reduced bleeding risk, and faster recovery compared to open abdominal approaches [83]. It also preserves the vaginal length unlike anterior colporrhaphy, and can be performed concurrently with other laparoscopic procedures, making it a potential first-line treatment for lateral defects in anterior vaginal wall prolapse [83].

### 5.3. Efficacy

Anterior colporrhaphy is effective in correcting anterior wall prolapse and improving symptoms. Long-term outcomes generally indicate good anatomical success and symptom relief, but some studies report varying results depending on factors such as the severity of prolapse and surgical technique [46,52]. Comparative studies suggest that paravaginal repair can offer superior outcomes compared to anterior colporrhaphy in cases of complex prolapse, especially with respect to addressing lateral support issues [29,83]. Regarding long-term efficacy, a retrospective study found comparable surgical outcomes between anterior colporrhaphy and paravaginal repair at 1–2 months, 3–6 months, and 1 year postoperatively, with paravaginal repair having longer operation times [29]. However, abdominal retropubic suspension appeared superior to anterior colporrhaphy in subjective cure rates and reduced need for repeat surgeries [92]. A long-term follow-up study comparing anterior colporrhaphy, anterior colporrhaphy with xenograft reinforcement, and mesh repair found good results for anterior colporrhaphy alone, with no significant increase in anatomic cure rates when using additional support materials [93].

### 5.4. Recurrence Rates

Recurrence rates in anterior colporrhaphy can vary, with some studies indicating a higher risk of recurrence compared to that via more complex repairs. Recurrence rates for anterior colporrhaphy typically range from 10% to 30%, depending on the initial prolapse severity and surgical technique [94]. Recurrence rates for paravaginal repair are generally lower than those for anterior colporrhaphy in cases with significant lateral defects [29]. However, there is still a risk of recurrence, particularly if the repair does not address all aspects of the prolapse or if patient adherence to postoperative care is insufficient [31,95].

### 5.5. Patient Selection

The ideal candidates for anterior colporrhaphy include those with isolated anterior wall prolapse without significant lateral defects or complex pelvic support issues [92]. The procedure is suitable for patients with less severe prolapse or those who have not previously undergone surgical correction. For paravaginal repair, ideal candidates are those with significant lateral vaginal wall detachment or recurrent prolapse where anterior colporrhaphy alone may not suffice [96]. This technique is also suitable for patients who have complex pelvic support issues or those requiring a more comprehensive repair. Complex pelvic support issue may include recurrent prolapse, where previous repairs have failed, or prolapse affecting multiple compartments (anterior, posterior, or apical) [97]. Significant lateral vaginal wall detachment occurs when the fascia connecting the vaginal wall to the pelvic sidewall weakens or detaches, disrupting the pelvic floor’s ability to support the bladder and other organs [90]. This leads to both lateral and anterior prolapse [10].

### 5.6. Complications and Morbidity

Potential complications associated with anterior colporrhaphy include infection, wound healing issues, postoperative pain, and the risk of recurrence [62]. Recovery time is generally shorter compared to that via more invasive procedures, but patients may still experience discomfort and require some downtime. Long-term morbidity is typically manageable with proper postoperative care. The complications of paravaginal repair include injury to surrounding structures (e.g., bladder, urethra, rectum), the recurrence of prolapse, and postoperative pain [83,88]. Recovery times can vary based on the approach used (open abdominal, laparoscopic, robotic, or transvaginal). Although minimally invasive techniques can offer faster recovery, the risk of complications may be higher in complex cases [83]. Overall morbidity may also be greater due to the complexity of the procedure and the potential for extended recovery periods.

Transvaginal mesh implantation for POP repair carries significant risks of complications [79]. A systematic review found mean complication rates of 27% for anterior, 20% for posterior, and 40% for combined mesh repairs, with grade ≥III complications occurring in 8%, 3.5%, and 13%, respectively [98]. Common complications include mesh erosion, voiding dysfunction, and dyspareunia [98]. Risk factors include surgical technique, surgeon experience, previous prolapse repair, concomitant hysterectomy, mesh properties, patient age, sexual activity, and smoking [98]. Vaginal exposure is the most frequent mesh-specific complication, often requiring partial or complete excision [99]. Despite these risks, one study reported 96.87% anatomical success and improved quality of life with transvaginal polypropylene mesh for anterior compartment prolapse [100]. However, careful patient selection, education, and risk factor mitigation are crucial before mesh implantation [98,99].

Due to vaginal mesh carrying significant risks of complications, the Independent Medicines and Medical Devices Safety Review recommended an immediate suspension of surgical mesh use for stress urinary incontinence in England in July 2018 [101]. This decision was prompted by reports of severe complications including debilitating pain and mobility issues from women who had undergone mesh surgery [101]. The pause was also extended to cover vaginal mesh for pelvic organ prolapse [102]. Prior to this, NICE had advised using mesh implants only as a last resort after non-surgical options were exhausted [103]. The suspension raised concerns about potential exposure to more complex procedures, with higher complication rates for stress urinary incontinence treatment [102]. In response to the Cumberlege review’s recommendations, England planned to establish specialist surgical mesh removal centers, set to open in April 2021, to address the challenges of mesh removal and provide specialized care for affected women [104].

The use of synthetic mesh for POP and stress urinary incontinence (SUI) repair has been controversial due to complications such as erosion, pain, and infections [105]. While transvaginal permanent mesh shows lower rates of prolapse awareness and recurrence compared to native tissue repair, it is associated with higher reoperation rates and complications [106]. Biologic mesh materials have been proposed as alternatives, showing similar clinical outcomes but lower adverse effects compared to synthetic meshes [105]. However, the evidence for absorbable mesh and biological grafts remains limited [106]. The debate surrounding mesh use in pelvic reconstructive surgery has become increasingly contentious [107]. Although open tension-free methods using mesh for hernia repair have been widely adopted, there has been little rigorous evaluation of these techniques compared to non-mesh methods [108].

In summary, the choice between anterior colporrhaphy and paravaginal repair depends on the specific characteristics of the prolapse, complexity of the case, and patient’s overall health. Each technique has its advantages and potential risks, with the decision typically guided by the nature of the vaginal wall defect and the desired surgical outcomes.

## 6. Innovations and Future Directions

### 6.1. Minimally Invasive Techniques

Minimally invasive techniques, such as laparoscopy and robotic surgery, have revolutionized the approach to pelvic floor repairs by reducing the need for large incisions, minimizing postoperative pain, and shortening recovery times [109]. Laparoscopy involves using small incisions and a camera to guide the surgeon in repairing pelvic organ prolapse or other defects [110]. It allows for precise maneuvers with less disruption to surrounding tissues, resulting in a quicker recovery and reduced postoperative pain [110]. Studies have shown that laparoscopic approaches can be as effective as traditional open surgeries with respect to outcomes, with the added benefits of shorter hospital stays and a faster return to normal activities [111]. Robotic-assisted surgery provides enhanced precision through the use of robotic arms controlled by the surgeon via a console [112]. This technology allows for greater dexterity and visualization, particularly in complex cases. Robotic surgery can lead to improved surgical outcomes including lower complication rates and faster recovery times. However, it also requires a significant investment in technology and training [113]. Training and certification for robotic procedures in pelvic organ prolapse require specialized surgical education, hands-on experience with robotic systems, and proficiency assessments to ensure that surgeons can perform these advanced techniques safely and effectively [114].

### 6.2. Use of Grafts and Mesh

The use of grafts and mesh materials in pelvic floor repair has been a significant advancement, aimed at reinforcing weakened tissues and reducing the risk of recurrence [115]. However, their use comes with both benefits and controversies. Grafts and mesh are often used in cases where traditional repairs might not provide sufficient support such as severe prolapse or recurrent defects [106]. They can be employed in both anterior colporrhaphy and paravaginal repair to provide additional structural reinforcement [79]. However, the use of synthetic mesh has been controversial due to concerns regarding complications such as infection, erosion, and chronic pain [79]. Some studies have reported higher rates of complications with mesh repairs than with non-mesh repairs, leading to debates about the safety and efficacy of these materials [79,80]. With respect to outcomes, mesh and grafts can offer effective long-term support and reduce recurrence rates when used appropriately [78]. The choice of material and technique, along with patient selection, plays a crucial role in determining the success of mesh-augmented repairs. Future research on pelvic organ prolapse repair could benefit from a closer examination of specific mesh types including synthetic mesh, which provides long-lasting support but carries risks like erosion; biological mesh, derived from human or animal tissue, which integrates naturally but may degrade over time; and absorbable mesh, designed to offer temporary support as the body heals, potentially reducing long-term complications but with a higher risk of recurrence [75,115,116]. Understanding the distinct advantages and limitations of each type will help optimize patient outcomes and guide advancements in surgical techniques.

### 6.3. Biomechanical Studies

Biomechanical studies focus on understanding the mechanical properties of pelvic tissues and how they respond to different repair techniques [117]. These studies are crucial for developing better surgical methods and materials. Research into the mechanical properties of pelvic tissues helps in designing more effective repair strategies that mimic natural tissue behavior and improve overall support [118]. This includes studying how tissues respond to stress and strain and how different repair materials interact with the surrounding structures [119]. Advances in biomechanical research are leading to the development of new materials and surgical techniques that enhance tissue support [120]. This includes the creation of biomaterials that better integrate with native tissue and the development of surgical techniques that reduce stress on repaired tissues [121,122]. Biomechanical studies face challenges like accurately modeling the complex, dynamic properties of human tissues, which vary between individuals and change over time [123]. Additionally, the reliance on laboratory models or simulations often falls short of capturing real-life conditions, underscoring the need for more in vivo research to validate findings and ensure that surgical innovations truly reflect the physiological behavior of human tissues during pelvic organ prolapse repair [124].

### 6.4. Role of Pelvic Floor Therapy

After POP surgery, patients are closely monitored through a series of follow-up appointments to assess healing, manage complications, and detect recurrence [125]. Typically, an initial postoperative visit occurs 4–6 weeks after surgery, during which wound healing is evaluated, and pelvic exams are performed to assess the stability of the repair and check for any signs of infection or mesh erosion if mesh was used. Patients are also asked about ongoing symptoms, such as pelvic pain, urinary incontinence, or constipation, which could indicate complications or issues with the repair [125]. In some cases, pelvic floor physical therapy is recommended to strengthen the pelvic muscles, promote healing, and help prevent recurrence [126]. If complications are suspected, imaging techniques such as ultrasound or MRI may be used to assess the pelvic floor [127,128]. After the initial recovery period, patients typically undergo annual follow-ups to monitor for recurrence or late-onset complications, ensuring long-term recovery and health.

Pelvic floor therapy plays a critical role in the overall management of pelvic floor disorders, both before and after surgical interventions. Preoperative pelvic floor therapy can help prepare the pelvic muscles for surgery, improve muscle strength, and educate patients on post-surgical care [129,130]. This can enhance surgical outcomes and reduce recovery times. Techniques such as pelvic floor exercises, biofeedback, and lifestyle modifications are commonly used [131]. Postoperative pelvic floor therapy focuses on rehabilitation to restore function and strength, prevent complications, and support recovery [129]. This may involve exercises to strengthen the pelvic muscles, techniques to manage pain and discomfort, and strategies to prevent the recurrence of prolapse [132]. Effective postoperative therapy is crucial for optimizing long-term outcomes and patient satisfaction [133].

In summary, innovations in minimally invasive techniques, the use of grafts and mesh, biomechanical studies, and pelvic floor therapy are shaping the future of pelvic floor repair. These advancements aim to enhance surgical outcomes, improve patient safety, and provide better overall care. As technology and research continue to evolve, these approaches are expected to further refine and improve the management of pelvic floor disorders.

## 7. Clinical Guidelines and Recommendations

### 7.1. Current Guidelines

The American Urological Association and the Society of Urodynamics, Female Pelvic Medicine and Urogenital Reconstruction provide guidelines for the management of pelvic organ prolapse including recommendations for anterior colporrhaphy and paravaginal repair [134]. Key recommendations include the following.

Anterior colporrhaphy is recommended for patients with uncomplicated anterior vaginal wall prolapse [135]. Guidelines emphasize the importance of individualized treatment plans based on prolapse severity and patient symptoms. Anterior colporrhaphy is typically suggested for patients with a simple anterior wall defect who may benefit from a less invasive procedure [46]. Paravaginal repair is recommended for patients with more complex prolapse involving lateral detachment or recurrent prolapse [84]. Guidelines highlight the use of this technique for addressing significant lateral defects or complex pelvic support issues [136]. It is considered a suitable option when anterior colporrhaphy alone is insufficient.

The Society of Urological Surgeons provides additional guidelines on the use of surgical mesh and grafts, emphasizing the importance of informed consent and patient education regarding the potential risks and benefits [137]. Recommendations stress the need for careful patient selection and the consideration of alternative repair options if mesh use is contraindicated. The National Institute for Health and Care Excellence offers guidelines that support minimally invasive techniques such as laparoscopy and robotic surgery where appropriate, with recommendations for ensuring that these approaches are used by experienced surgeons and in suitable cases [138].

### 7.2. Decision Making in Practice

During risk assessment and deciding between anterior colporrhaphy and paravaginal repair, it is crucial to assess the individual risks associated with each procedure. Anterior colporrhaphy generally carries a lower risk profile with a less invasive approach, but it may not be suitable for complex prolapse cases [139]. Paravaginal repair, although effective for more severe or recurrent prolapse, carries risks related to its complexity including potential injury to surrounding structures and higher complication rates [83]. With respect to outcome expectations, the anticipated outcomes of each procedure must be considered. Anterior colporrhaphy is effective for simple anterior wall defects, with good short-term results. However, it may also have higher recurrence rates for complex cases [46,52]. Paravaginal repair can offer more durable results for complex prolapse but may also involve longer recovery and a higher risk of complications [83].

Patient preferences play a critical role in decision making [140]. Factors such as the patient’s desire for minimally invasive options, their comfort with potential recovery times, and their priorities regarding symptom relief and quality of life should be considered [141]. Engaging patients in discussions about the benefits and risks of each procedure helps ensure that their preferences are aligned with the chosen surgical approach [142]. The surgeon’s expertise with each technique also affects decision making [143]. Choosing a procedure that aligns with the surgeon’s experience and skill set is essential for optimizing outcomes and minimizing complications [140]. Furthermore, providing patients with comprehensive information about their options including potential risks, benefits, and recovery expectations is crucial [144]. This process helps patients make informed decisions that align with their health goals and personal preferences.

In summary, clinical guidelines for anterior colporrhaphy and paravaginal repair emphasize tailored treatment approaches based on the severity of prolapse and patient characteristics. Decision making in clinical practice involves balancing risks, expected outcomes, and patient preferences to select the most appropriate surgical intervention.

## 8. Conclusions

Anterior colporrhaphy remains a cornerstone in the management of anterior vaginal wall prolapse. It is favored for its relative simplicity and effectiveness in addressing uncomplicated anterior wall defects. Anterior colporrhaphy is associated with a lower risk profile and is typically chosen for patients with less complicated prolapse. However, despite its efficacy, recurrence can be a concern, particularly in more complex cases or if additional pelvic support issues are present. Paravaginal repair is increasingly recognized for its usefulness in addressing more complex prolapse cases including significant lateral detachment of the vaginal wall and recurrent prolapse. Paravaginal repair is particularly valuable in cases where anterior colporrhaphy alone may be insufficient. It offers robust support for severe or recurrent defects, although it is associated with increased complexity and potential risks such as injury to surrounding structures and longer recovery times. The comparison between these procedures is summarized in Table 1.

### 8.1. Future Research and Innovation Areas

#### 8.1.1. Advancements in Minimally Invasive Techniques

Continued development and refinement of laparoscopic and robotic surgical techniques are crucial [145]. Current research is focused on improving precision, reducing complications, and further shortening recovery times. Innovations in imaging and surgical tools can enhance the effectiveness of minimally invasive repairs and expand their applicability to more complex cases.

#### 8.1.2. Improvement and Innovation in Mesh and Graft Materials

Research is needed to develop safer, more effective mesh and graft materials that minimize complications such as erosion and infection [146]. Advances in biocompatible and absorbable materials are expected to improve patient outcomes and reduce the risks associated with synthetic meshes. Ongoing studies aim to identify optimal materials and techniques for incorporating grafts and meshes into surgical repairs.

#### 8.1.3. Biomechanical Studies for Enhanced Tissue Repair

Further biomechanical research is needed to better understand the mechanical properties of pelvic tissues and the mechanisms by which different repair methods and materials interact with these tissues [123]. This knowledge will help in designing more effective repair strategies and materials that better mimic natural tissue behavior and provide long-term support.

#### 8.1.4. Integration of Pelvic Floor Therapy

Future research should explore the optimal integration of pelvic floor therapy into pre- and postoperative care [147]. Studies focusing on the impact of pelvic floor exercises, biofeedback, and other therapeutic modalities on surgical outcomes and recovery will be valuable in developing comprehensive management strategies for pelvic floor disorders.

#### 8.1.5. Long-Term Outcomes and Comparative Effectiveness Research

There is a need for more long-term, comparative effectiveness research to evaluate the outcomes of different surgical techniques and materials [69]. Such studies will provide valuable insights into the relative benefits and risks of anterior colporrhaphy in comparison with those of paravaginal repair, helping guide clinical decision making and improve patient outcomes.

In summary, both anterior colporrhaphy and paravaginal repair play important roles in managing pelvic organ prolapse, with each technique offering distinct advantages depending on the complexity of the prolapse (Figure 1). Future research and innovations are likely to focus on enhancing minimally invasive techniques, improving repair materials, understanding tissue biomechanics, and integrating comprehensive pelvic floor therapy to optimize surgical outcomes and patient care.

## Figures and Tables

**Figure 1 medicina-60-01865-f001:**
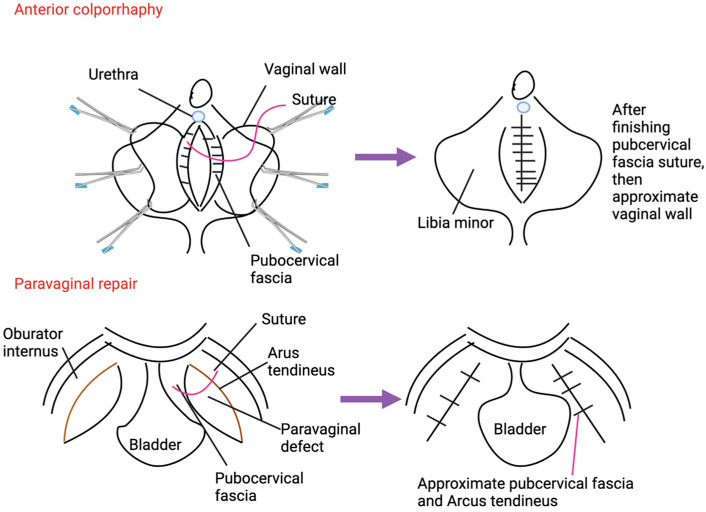
Comparison between anterior colporrhaphy and paravaginal repair regarding anatomy and techniques.

**Table 1 medicina-60-01865-t001:** Comparison between anterior colporrhaphy and paravaginal repair.

Aspect	Anterior Colporrhaphy	Paravaginal Repair
Indication	Simple anterior vaginal wall prolapse (cystocele)	Lateral vaginal wall detachment or complex prolapse
Approach	Transvaginal approach (most common)	Can be performed transvaginally, laparoscopically, or abdominally
Procedure	Cut and suturing the weakened anterior vaginal wall to restore support	Reattachment of the detached lateral vaginal wall (pubocervical fascia) to the pelvic sidewall (ATFP)
Complexity	Less complex, straightforward	More complex, involves repairing lateral detachment
Duration of surgery	Typically shorter	Longer due to complexity
Hospital stay	Shorter (often outpatient)	May require a longer hospital stay, especially for laparoscopic or abdominal approaches
Recurrence rate	Higher recurrence in complex prolapse cases	Lower recurrence rate, especially for lateral defects
Efficacy	Effective for mild to moderate prolapse	More effective for severe or recurrent prolapse
Recovery time	Quicker recovery (usually 4–6 weeks)	Longer recovery of usually 6–8 weeks, depending on the approach
Use of mesh/grafts	Usually not required	May involve mesh or grafts to reinforce the repair
Complication risk	Lower risk of complications	Higher risk due to the complexity and proximity to important structures (e.g., bladder, ureters)
Patient suitability	Suitable for younger, healthier patients with mild prolapse	Recommended for patients with significant lateral defects or recurrence
Long-term success	Good short-term success but higher recurrence in complex cases	Good long-term outcomes with lower recurrence rates
Pelvic floor impact	May result in weakening over time due to lack of lateral support	Provides stronger lateral and central pelvic floor support

## Data Availability

Not applicable.

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
