# Peer review of "Anterior Colporrhaphy and Paravaginal Repair for Anterior Compartment Prolapse: A Review"

_medicina, 2024, doi:10.3390/medicina60111865_

Round 1

Reviewer 1 Report

Comments and Suggestions for Authors

1. Add a concise explanation of the normal anatomy and function of the pelvic floor and its role in maintaining organ support. This will provide a stronger foundation for discussing prolapse.

2. Include data on how common anterior compartment prolapse is, especially in comparison to other types of prolapse. Mention rates of surgical intervention if relevant.

3. Add a concluding sentence to the introduction that introduces the focus on surgical management, especially anterior colporrhaphy and paravaginal repair, and briefly mention why these approaches are important in managing anterior prolapse.

4. Further explore how symptoms like urinary incontinence, recurrent infections, and sexual dysfunction can affect daily life and why surgical treatment might be necessary when conservative measures fail.

5.  Clarify the mechanism of action in a more detailed manner. For instance, you could briefly explain how plicating or suturing the weakened fascia restores the anteroposterior support of the bladder and prevents its descent.

6. Provide more clinical context for why this procedure is often required despite conservative treatments. For example, you can mention specific situations, such as the degree of prolapse or the failure of symptom relief despite the use of pessaries and other non-invasive methods.

7. Clarify the benefits and risks associated with modified techniques. In addition, highlight specific types of graft materials (biologic vs. synthetic) and their respective advantages and drawbacks.

8. Highlight the factors that most influence recurrence, such as age, degree of prolapse, and postoperative tissue quality. A more nuanced discussion about the long-term durability of the procedure and factors affecting recurrence would provide further insights.

9.You could expand on the types of stress urinary incontinence (e.g., urgency incontinence, mixed incontinence) that may emerge postoperatively. Additionally, mention the impact of these complications on the patient's long-term quality of life.

10. Emphasize the postoperative recovery phase, the need for patient education, and strategies to improve outcomes. This could include postoperative pelvic floor rehabilitation and the role of follow-up care.

11. Briefly provide the rationale behind the FDA's decision and its impact on clinical practice, while also mentioning alternatives to mesh that have gained favor.

12. Summarize the long-term effectiveness of anterior colporrhaphy and emphasize the need for tailored approaches based on patient-specific factors.

13. regarding Surgical Techniques include a more detailed comparison of techniques, especially in terms of their advantages, disadvantages, and outcomes (i.e., risks and benefits of each approach).

14. Provide a clear concluding statement summarizing when paravaginal repair should be considered over other approaches and its long-term role in the management of pelvic organ prolapse.

15. regarding 

  • "Anterior colporrhaphy is effective in correcting anterior wall prolapse and improving symptoms."

  • "Paravaginal repair... often provides high rates of anatomical correction and symptom improvement."

  • Add specific comparisons of efficacy between the two approaches:
    • You might consider explicitly stating that paravaginal repair has shown superior anatomical outcomes in cases of lateral vaginal wall defects.
    • The comparison could also briefly address long-term efficacy and patient-reported outcomes.
16. The patient selection criteria for anterior colporrhaphy and paravaginal repair are mentioned, but further clarification could improve the understanding of what specifically constitutes “complex pelvic support issues” or “significant lateral vaginal wall detachment.”

17. Since the use of advanced technology like robotics is mentioned, it would be helpful to briefly note the requirements for training and certification to ensure the quality and safety of these procedures.

18. A brief mention of specific types of mesh (e.g., synthetic vs. biological vs. absorbable) could provide additional clarity and direction for future research

19. Mention challenges in biomechanical studies, such as the difficulty in modeling human tissues accurately and the need for more in-vivo research to confirm biomechanical findings.

20. in the postoperative managements address postoperative follow-ups more explicitly in this section. How are patients monitored after surgery? What are the common follow-up protocols to manage potential recurrence or complications?

  •  

Author Response

Reviewer 1

Q1. Add a concise explanation of the normal anatomy and function of the pelvic floor and its role in maintaining organ support. This will provide a stronger foundation for discussing prolapse.

Response 1: We thank the reviewer’s comment. We have added Anatomy and function of the pelvic floor in the 2nd paragraph of introduction. (Setion 1, page 1, lines 33-42).  

Q2. Include data on how common anterior compartment prolapse is, especially in comparison to other types of prolapse. Mention rates of surgical intervention if relevant.

Response 2: We thank the reviewer’s comment. We have added “Prevalence of anteiror vaginal wall prolapse” in the last 3rd paragraph of introduction. (Section 1, page 2. lines 81-90). 

Q3. Add a concluding sentence to the introduction that introduces the focus on surgical management, especially anterior colporrhaphy and paravaginal repair, and briefly mention why these approaches are important in managing anterior prolapse.

Response 3: We thank the reviewer’s comment. We have added a concluding sentence regarding the focus on surgical management and the importance in managing anterior prolapse. (Section 1, page 2, lines 93-97)

Q4. Further explore how symptoms like urinary incontinence, recurrent infections, and sexual dysfunction can affect daily life and why surgical treatment might be necessary when conservative measures fail.

Response 4: We thank the reviewer’s comment. We have elaborate the how the POP symptoms affect daily life and why surgical treatment might be necessary when conservative measures fail. (Section 1, page 2. lines 69-75)

Q5.  Clarify the mechanism of action in a more detailed manner. For instance, you could briefly explain how plicating or suturing the weakened fascia restores the anteroposterior support of the bladder and prevents its descent.

Response 5: We thank the reviewer’s comment. We have added a detailed mechanism regarding how plicating or suturing the weakened fascia restores the anteroposterior bladder support and prevents its descent. (Section 3.4, page 4, lines 182-194)

Q6. Provide more clinical context for why this procedure is often required despite conservative treatments. For example, you can mention specific situations, such as the degree of prolapse or the failure of symptom relief despite the use of pessaries and other non-invasive methods.

Response 6: We thank the reviewer’s comment. We have added a paragraph to discuss the point. (Section 3.5, page 4, lines 196-209)

Q7. Clarify the benefits and risks associated with modified techniques. In addition, highlight specific types of graft materials (biologic vs. synthetic) and their respective advantages and drawbacks.

Response 7: We thank the reviewer’s comment. We have added a paragraph to discuss the point. (Section 3.8, page 6, lines 281-292) 

Q8. Highlight the factors that most influence recurrence, such as age, degree of prolapse, and postoperative tissue quality. A more nuanced discussion about the long-term durability of the procedure and factors affecting recurrence would provide further insights.

Response 8: We thank the reviewer’s comment. We have added a paragraph to discuss the point. (Section 3.5, page 5, lines 215-226)

Q9.You could expand on the types of stress urinary incontinence (e.g., urgency incontinence, mixed incontinence) that may emerge postoperatively. Additionally, mention the impact of these complications on the patient's long-term quality of life.

Response 9: We thank the reviewer’s comment. We have added a paragraph to discuss the point. (Section 3.7, page 5, lines 246-258)

Q10. Emphasize the postoperative recovery phase, the need for patient education, and strategies to improve outcomes. This could include postoperative pelvic floor rehabilitation and the role of follow-up care.

Response 10: We thank the reviewer’s comment. We have added a paragraph to discuss the point. (Section 3.7, page 6, lines 263-274)

Q11. Briefly provide the rationale behind the FDA's decision and its impact on clinical practice, while also mentioning alternatives to mesh that have gained favor.

Response 11: We thank the reviewer’s comment. We have expanded the complication section and mentioned alternatives to mesh. (Section. 5.5. page 9, lines 449-460)

Q12. Summarize the long-term effectiveness of anterior colporrhaphy and emphasize the need for tailored approaches based on patient-specific factors.

Response 12:  We thank the reviewer’s comment. We have added a paragraph to summarize the long-term effectiveness of anterior colporrhaphy and the need for tailored approaches based on patient-specific factors. (Section 3.6, page 5)

Q13. regarding Surgical Techniques include a more detailed comparison of techniques, especially in terms of their advantages, disadvantages, and outcomes (i.e., risks and benefits of each approach).

Response 13: We thank the reviewer’s comment. We have added a paragraph to compare their techniques and outcomes. (Section 5.2, page 8)

Q14. Provide a clear concluding statement summarizing when paravaginal repair should be considered over other approaches and its long-term role in the management of pelvic organ prolapse.

Response 14: We thank the reviewer’s comment. We have added a paragraph to conclude when paravaginal repair should be a better choice for anterior wall prolapse. (Section 5.2, page 8, lines 378-382)

Q15. regarding 

  • "Anterior colporrhaphy is effective in correcting anterior wall prolapse and improving symptoms."
  • "Paravaginal repair... often provides high rates of anatomical correction and symptom improvement."
  • Add specific comparisons of efficacy between the two approaches:
    • You might consider explicitly stating that paravaginal repair has shown superior anatomical outcomes in cases of lateral vaginal wall defects.
    • The comparison could also briefly address long-term efficacy and patient-reported outcomes.

Response 15: We thank the reviewer’s comment. We have rewritten the sentence as the reviewer suggested and we have also briefly address long-term efficacy of both surgeries. (Section 5.2, page 8, lines 389-390, 391-399) 

Q16. The patient selection criteria for anterior colporrhaphy and paravaginal repair are mentioned, but further clarification could improve the understanding of what specifically constitutes “complex pelvic support issues” or “significant lateral vaginal wall detachment.”

Response 16: We thank the reviewer’s comment. We have added a paragraph further clarification complex pelvic support issues and significant lateral vaginal wall detachment. (Section 5.4, page 8, lines 418-423)

Q17. Since the use of advanced technology like robotics is mentioned, it would be helpful to briefly note the requirements for training and certification to ensure the quality and safety of these procedures.

Response 17: We thank the reviewer’s comment. We have added a sentence to describe the requirements for training and certification to ensure the quality and safety of these procedures. (Section 6.1, page 10, lines 492-495)

  1. A brief mention of specific types of mesh (e.g., synthetic vs. biological vs. absorbable) could provide additional clarity and direction for future research

Response 18: We thank the reviewer’s comment. We have added a paragraph to describe clarity and direction for future research for different types of mesh. (page 10, lines 496-504)

Q19. Mention challenges in biomechanical studies, such as the difficulty in modeling human tissues accurately and the need for more in-vivo research to confirm biomechanical findings.

Response 19: We thank the reviewer’s comment. We have added a paragraph to describe challenge in biomechanical studies. (Section 6.2, page 10, lines 511-518)

Q20. in the postoperative managements address postoperative follow-ups more explicitly in this section. How are patients monitored after surgery? What are the common follow-up protocols to manage potential recurrence or complications?

Response 20: We thank the reviewer’s comment. We have added a paragraph to describe postoperative follow-ups. (Section 6.4, page 10, lines 539-551)

Reviewer 2 Report

Comments and Suggestions for Authors

Strengths of the manuscript include:

·        Clear focus - a comparison of two techniques: anterior colporrhaphy and paravaginal repair techniques. It presents aspects such as indications, surgical approaches, outcomes, and complications.

·        Important topic coverage: You’ve covered key aspects, including anatomy, techniques, recurrence rates, and future directions, which make the manuscript informative. However, the surgical techniques are only briefly described in the review and potential intraoperative challenges, particular for each approach, are not even mentioned.

Areas that require improvement:

·       The introduction could benefit from clearer objectives. While you discuss POP in general, consider narrowing the focus to set the stage for the comparison between the two procedures.

·        Adding prevalence data and also data on the impact of the economic  burden of this pathology at a societal level should be considered.

·        You could define the main challenge earlier—whether it's high recurrence rates or difficulty in selecting the appropriate surgical approach.

·        Multiple repetitions of the definition of anterior compartment prolapse are found within the introduction section, as well as the rest of the manuscript.

·        Any inclusion / exclusion criteria for study selection have not been mentioned. Adding these criteria would increase the transparency of your review and allow readers to access up-to-date information.

·        I suggest expanding the section on complications, particularly the implications of mesh-related complications that have been highly scrutinized recently. Since this is a hot issue in pelvic surgery, diving deeper into the risks associated with each technique would make the manuscript more comprehensive.

·        The Independent Medicines and Medical Devices Safety Review recommended a pause on the use of vaginal mesh for SUI which was accepted by the UK government and NHS, and implemented in July 2018. The pause was subsequently extended to cover the use of vaginal mesh for POP. Whilst the meshes are not currently banned, they are only used as a last resort through a high vigilance programme of restricted practice. Nothing was mentioned in the manuscript regarding this important topic in European practice.

·        The table comparison between anterior colporrhaphy and paravaginal repair is suitable, but you might consider including diagrams or surgical illustrations to visually support the descriptions of anatomy and technique.

·        Figure 1 does not bring any new information; it just repeats the information mentioned above in the text.

By addressing the mentioned issues, your manuscript will become more impactful.

Author Response

Reviewer 2 

Strengths of the manuscript include:

  •       Clear focus - a comparison of two techniques: anterior colporrhaphy and paravaginal repair techniques. It presents aspects such as indications, surgical approaches, outcomes, and complications.
  •       Important topic coverage: You’ve covered key aspects, including anatomy, techniques, recurrence rates, and future directions, which make the manuscript informative. However, the surgical techniques are only briefly described in the review and potential intraoperative challenges, particular for each approach, are not even mentioned.

Areas that require improvement:

Q1. The introduction could benefit from clearer objectives. While you discuss POP in general, consider narrowing the focus to set the stage for the comparison between the two procedures.

Response 1: We thank the reviewer’s comment. We have added a paragraph focus on anatomy of the pelvic floor. (Section 1, page 2, lines 43-54)  

Q2.  Adding prevalence data and also data on the impact of the economic  burden of this pathology at a societal level should be considered.

Response 2: We thank the reviewer’s comment. We added a paragraph to discuss the prevalence and economic burden of the POP. (Section 1, page 1, lines 81-89)

Q3. You could define the main challenge earlier—whether it's high recurrence rates or difficulty in selecting the appropriate surgical approach.

Response 3: We thank the reviewer’s comment. We added a paragraph to discuss main challenge of the two surgical procedure in the last paragraph of introduction. (Section 1, page 3, lines 112-116)

Q4.  Multiple repetitions of the definition of anterior compartment prolapse are found within the introduction section, as well as the rest of the manuscript.

Response 4: We thank the reviewer’s comment. We have deleted the repeated definition of anterior compartment prolapse in the maintext. 

Q5.  Any inclusion / exclusion criteria for study selection have not been mentioned. Adding these criteria would increase the transparency of your review and allow readers to access up-to-date information.

Response 5: We thank the reviewer’s comment. We have provided the inclusion and exclusion criteria for our study in section 1.2 (Section 1.2, page 3, lines 118-125) 

Q6. I suggest expanding the section on complications, particularly the implications of mesh-related complications that have been highly scrutinized recently. Since this is a hot issue in pelvic surgery, diving deeper into the risks associated with each technique would make the manuscript more comprehensive.

Response 6: We thank the reviewer’s comment. We have extended compilation section as suggested. (Section 5.5, page 9, lines 437-448)

Q7. The Independent Medicines and Medical Devices Safety Review recommended a pause on the use of vaginal mesh for SUI which was accepted by the UK government and NHS, and implemented in July 2018. The pause was subsequently extended to cover the use of vaginal mesh for POP. Whilst the meshes are not currently banned, they are only used as a last resort through a high vigilance programme of restricted practice. Nothing was mentioned in the manuscript regarding this important topic in European practice.

Response 7: We thank the reviewer’s comment. We added this part in the section. (Section 5.5, page 9, lines 449-460)

Q8. The table comparison between anterior colporrhaphy and paravaginal repair is suitable, but you might consider including diagrams or surgical illustrations to visually support the descriptions of anatomy and technique.

Response 8: We thank the reviewer’s comment. We added a diagram to compare the anatomy and technique of both procedures (Figure 1).  

Q9.  Figure 1 does not bring any new information; it just repeats the information mentioned above in the text.

Response 9: We thank the reviewer’s comment. We have deleted the original Figure 1.

By addressing the mentioned issues, your manuscript will become more impactful.